# Induced Mutation in *GmCOP1b* Enhances the Performance of Soybean under Dense Planting Conditions

**DOI:** 10.3390/ijms23105394

**Published:** 2022-05-12

**Authors:** Ronghuan Ji, Xinying Xu, Jun Liu, Tao Zhao, Hongyu Li, Jixian Zhai, Bin Liu

**Affiliations:** 1The National Key Facility for Crop Gene Resources and Genetic Improvement (NFCRI), Institute of Crop Science, Chinese Academy of Agricultural Sciences, Beijing 100081, China; jironghuan@sina.com (R.J.); xuxinying1004@sina.com (X.X.); liujun02@caas.cn (J.L.); zhaotao02@caas.cn (T.Z.); lihongyu@caas.cn (H.L.); 2Department of Biology, Southern University of Science and Technology, Shenzhen 518055, China; zhaijx@sustech.edu.cn

**Keywords:** soybean, photomorphogenesis, shade avoidance, stem elongation

## Abstract

CONSTITUTIVE PHOTOMORPHOGENIC 1 (COP1) is the key photomorphogenic inhibitor that has been extensively studied in higher plants. Nevertheless, its role has not been documented in the economically important soybean. Here we investigated the functions of two *COP1* homologous genes, *GmCOP1a* and *GmCOP1b*, by analyzing *Gmcop1a* and *Gmcop1b* mutants with indels using CRISPR in soybean. We revealed that, although both genes are required for skotomorphogenesis in the dark, the *GmCOP1b* gene seems to play a more prominent role than *GmCOP1a* in promoting stem elongation under normal light conditions. Consistently, the bZIP transcriptional factors STF1/2, which repress stem elongation in soybean, accumulated to the highest level in the *Gmcop1a1b* double mutant, followed by the *Gmcop1b* and *Gmcop1a* mutants. Furthermore, the *Gmcop1b* mutants showed reduced shade response and enhanced performance under high-density conditions in field trials. Taken together, this study provides essential genetic resources for elucidating functional mechanisms of GmCOP1 and breeding of high yield soybean cultivars for future sustainable agriculture.

## 1. Introduction

Soybean (*Glycine max* (L.) Merr.), as an important legume crop, provides worldwide human food, animal feed, and industrial raw materials [1]. Statistically, soybean accounts for 59% of the oilseed production and 70% of protein meal consumption in the world (http://www.soystats.com/, accessed on 9 January 2022). With the continuous growth of the world population and the improvement of living standards, the total production of soybean must increase by about 2.4% per year to meet the increasing demand for soybean [2,3]. In contrast to rice and wheat, whose yields have remarkably increased through the ‘Green Revolution’ by utilizing semi-dwarf varieties carrying allelic mutations in the genes of the Gibberellin (GA) pathway [4,5,6], soybean has not experienced such innovations to achieve a fundamental boosting of yield. It has been proposed that an ideotype of soybean for its imminent ‘Green Revolution’ should be characterized by a strong stem with shorter internode lengths and an appropriate plant height with more node numbers, which is lodging resistant and suitable for density planting [7].

Plant height is one of the most important agronomic traits determined by node number and internode length in soybean. Recent genetic studies have identified a few plant-height regulating genes in soybean, including *Dt1* (indeterminate habit), *Dt2* (semi-determinate habit), *GmDW1* (dwarf mutant), *LATE ELONGATED HYPOCOTYL* (*GmLHY*), *APETALA1* (*GmAP1*), and *CRYPTOCHROME1* (*GmCRY1*). The *Dt1* gene is an ortholog of Arabidopsis *TERMINAL FLOWER 1* (*TFL1*), which inhibits the transition from vegetative to reproductive shoot apical meristem in soybean [8,9]. *Dt2* is a MADS family gene homologous to Arabidopsis *APETALA1/SQUAMOSA* (*AP1/SQUA*), which represses the expression of *Dt1* in the shoot apical meristem and promotes flora initiation [9,10]. Thus, *Dt1* and *Dt2* coordinately regulate plant height by affecting the growth habits and node numbers in soybean. The *GmDW1* gene encodes an *ent*-Kaurene synthase required for the biosynthesis of gibberellins (GA) to promote stem elongation [11]. The four *GmLHY* genes and the four *GmAP1* genes function as enhancers and repressors of internode elongation, respectively, possibly by altering the endogenous GA content [12,13]. Our previous study revealed that the blue light receptor GmCRY1s stabilizes the bZIP transcription factors STF1 and STF2 to enhance the expression of GA2 oxidases and thus meditates light repression of internode elongation in soybean [14].

Light, as one of the most important environmental factors, not only provides the energy for photosynthesis but also serves as the signal regulating plant growth and development during the entire life cycle [15,16,17]. Arabidopsis seedlings undergo skotomorphogenic growth characterized by elongated hypocotyls and apical hooks in the dark and display a photomorphogenic phenotype including short hypocotyls and open cotyledons under light conditions [17,18,19]. The *CONSTITUTIVE PHOTOMORPHOGENIC 1* (*COP1*) gene is the key regulator repressing photomorphogenesis in the absence of light. The *cop1* mutants exhibit constant photomorphogenic phenotypes even in the continuous dark [20]. COP1 consists of three domains, the N-terminal RING domain, the middle coiled-coil domain, and the C-terminal WD40 domain [21,22,23]. In the dark, COP1 promotes the ubiquitination and degradation of transcriptional factors (TFs), including ELONGATED HYPOCOTYL 5 (HY5) [23], HY5-HOMOLOG (HYH) [24], LONG AFTER FAR-RED LIGHT 1 (LAF1) [25], and LONG HYPOCOTYL IN FAR RED (HFR1) [26], to repress photomorphogenesis. Under red or blue light, PHYTOCHROME A (PHYA), PHYTOCHROME B (PHYB), CRYPTOCHROME 1 (CRY1), and CRYPTOCHROME 2 (CRY2) stabilize above photomorphogenic TFs by suppressing the COP1 activity [27,28,29,30]. Under ultraviolet light (UV), the UVB light receptor UV RESISTANCE LOCUS8 (UVR8) binds to COP1 and transforms COP1 from a negative regulator of photomorphogenesis to a positive regulator by stabilizing HY5 [31].

In addition to the studies in Arabidopsis, the functions of COP1 in light signaling have been investigated in crops including pea [32], tomato [33], and rice [34,35,36,37]. The pea *light-independent photomorphogenesis1* (*lip1*) mutant harboring a mutation in the *COP1* homologous gene showed short stems and expanded shoots when grown in darkness [32]. Overexpression of *Solanum melongena COP1* (*SmCOP1*) could inhibit fruit ripening by suppressing ethylene synthesis in tomato fruit [33]. *OsCOP1* could restore the skotomorphogenesis phenotype of Arabidopsis *cop1* and is involved in regulating flavonoid synthesis and embryo development in rice [36]. However, the role of COP1 in soybean remains unclear. Our previous studies showed that overexpression of *GmCRY1b* genes significantly enhanced the lodging resistant ability and increased the yield potential of soybean under dense farming conditions [14]. We surmised that knockout of *COP1* homologous gene in soybean might also confer similar effects as overexpression of *GmCRY1b*. In this study, we knocked out the two *COP1* homologous genes, *GmCOP1a* and *GmCOP1b,* through the CRISPR/Cas9 technology. We confirmed that *GmCOP1a* and *GmCOP1b* are functionally redundant in the repression of photomorphogenesis in the dark, and both of them are involved in the shade avoidance response induced by low blue light. Notably, the *Gmcop1b* mutants were semi-dwarf and lodging resistant in comparison to WT, which are suited for planting under dense planting conditions to achieve a higher yield.

## 2. Results

### 2.1. The COP1 Homologous Proteins in Soybean

To investigate the possible roles of COP1 in soybean light response, we searched the soybean genomic sequence database by using the protein sequence Arabidopsis COP1 and identified two soybean *COP1*-like genes, *GmCOP1a* (*Glyma.02G267800*) and *GmCOP1b* (*Glyma.14G049700*). Phylogenetic analysis showed that plant and animal COP1 proteins were grouped into different branches, with the plant branch further divided into the monocotyledon and dicotyledon subbranches (Figure 1A). The two GmCOP1s were grouped together with the Arabidopsis COP1 (AtCOP1) in the dicotyledon subbranch. Protein alignment results showed that GmCOP1s have a high similarity with AtCOP1 in Arabidopsis (Appendix A Appendix A). They all have a RING domain at the N-terminus, a coiled-coil in the middle, and seven WD40 repeated domains at the C-terminus (Appendix A Appendix A). The identity between GmCOP1a and GmCOP1b is 95.6%, and their identities with AtCOP1 are 75.8% and 77.3%, respectively, suggesting that GmCOP1s are functionally conserved with AtCOP1. To test this speculation, we made the *YFP-GmCOP1a* (YFP, yellow fluorescent protein) and *YFP-GmCOP1b* constructs and transformed them into Arabidopsis *cop1–4* mutant, which is characterized by short hypocotyl, open cotyledon, and other continuous photomorphogenesis phenotypes in the dark. The result showed that ectopic expression of individual *YFP-GmCOP1a* and *YFP-GmCOP1b* could rescue the *cop1–4* mutant to different extents, conferring skotomorphogenesis phenotypes including elongated hypocotyl and apical hook when grown under continuous dark condition (Figure 1B–D). The hypocotyl lengths of respective transgenic lines were positively correlated with the abundance of the transgenic proteins (Appendix A Appendix A). The *YFP-GmCOP1a* and *YFP-GmCOP1b* ectopic expression also partially rescued the dwarf phenotype of the *cop1–4* mutant at the adult vegetative stage (Appendix A Appendix A). Although the transgenes resulted in enlarged leaf size, the leaf shape was as round as that of the *cop1–4* mutant. We speculate that this may be due to the different expression patterns of the 35S promoter and the endogenous promoter or due to functional differences between GmCOP1s and Arabidopsis COP1. Taken together, the above results suggest that the GmCOP1s have conserved functions as AtCOP1 in regulating plant growth and development.

### 2.2. The Expression Profiles of GmCOP1a and GmCOP1b

Next, we tested the transcript levels of *GmCOP1s* in different tissues. The result showed that both *GmCOP1a* and *GmCOP1b* have a higher expression level in cotyledon, unifoliolate leaf, and trifoliolate leaf, followed by the stem, apex, but a lower level in the root (Figure 2), implying that GmCOP1s majorly function in the light-perceiving aerial tissues. The expression patterns of *GmCOP1a* and *GmCOP1b* are very similar, suggesting that they have redundant functions in soybean. The overall expression levels of *GmCOP1b* in different tissues were moderately higher than that of *GmCOP1a* (Figure 2), implying that the *GmCOP1b* gene may play a relatively robust role in regulating soybean morphogenesis.

### 2.3. Redundant Roles of GmCOP1a and GmCOP1b in Skotomorphogenesis

To dissect the function of GmCOP1s in soybean, we knocked out *GmCOP1a* and *GmCOP1b* by the CRISPR/Cas9 technology and genetic transformation. Two gRNAs targeting the exons of each gene were designed through the CRISPR-P website (Figure 3A). We obtained three independent homozygous mutants for each gene (including *Gmcop1a-1, 2, 3* and *Gmcop1b-1, 2, 3*), which harbored premature mutation around their respective gRNA targeting sites (Figure 3B,C). The homozygous *Gmcop1a-2/1b-2* double mutant was further obtained by crossing the *Gmcop1a-2* and *Gmcop1b-2* mutants. We planted all the above mutant lines together with the wild-type TL1 in the dark for 5 days. The phenotypic result showed that the *Gmcop1a-2/1b-2* double mutant showed constitutive photomorphogenic phenotypes, including short hypocotyls and open cotyledons (Figure 3D,E). The *Gmcop1a* and *Gmcop1b* single mutants behaved similarly to the wild-type TL1, characterized by elongated hypocotyls and apical hooks (Figure 3D,E). The above results demonstrate that GmCOP1a and GmCOP1b are functionally redundant in regulating soybean skotomorphogenesis in the dark.

### 2.4. GmCOP1a and GmCOP1b Regulate Plant Height in Soybean

To test if the absence of GmCOP1a or GmCOP1b affects soybean growth and development in response to light, we planted the indicated lines under short- or long-day conditions for 10 days. The phenotypic analysis revealed that the *Gmcop1b* mutants, but not the *Gmcop1a* mutants, showed shorter epicotyl and hypocotyl compared to the wild-type TL1 (Figure 4A,B and Appendix A), supporting that GmCOP1b plays a more prominent role than GmCOP1a in promoting stem elongation under light conditions. The *Gmcop1a-2/1b-2* double mutant showed extremely dwarf status with much shorter hypocotyl and epicotyl than the *Gmcop1b* mutants, indicating that GmCOP1a is additive/redundant with GmCOP1b in regulating stem elongation (Figure 4A,B and Appendix A).

Next, we tried to get insight into the mechanisms of how GmCOP1s regulate stem elongation in response to light. Given that Arabidopsis COP1 inhibits photomorphogenesis through destabilizing HY5 in the dark, we tested if GmCOP1s also regulate the abundance of STF1/2, the homologs of Arabidopsis HY5. We examined the levels of STF1/2 in the unifoliate leaves of different genotypes subjected to dark–light transition. Quantitative immunoblot analysis showed that STF1/2 accumulated at equally low levels in the wild-type TL1, *Gmcop1a* and *Gmcop1b* mutants kept in the dark for 24 h. The light treatment gradually increased the abundance of STF1/2 with higher efficiency in the *Gmcop1b* mutant than in TL1 and the *Gmcop1a* mutants. STF1/2 in the *Gmcop1a-2/1b-2* double mutant were about four times more abundant than in wild-type TL1 in the dark but insensitive to dark–light transition (Figure 4C), which is consistent with its constitutive photomorphogenesis phenotype in the dark (Figure 3D,E). Taking the above results together with the fact that STF1/2 mediate light repression of stem elongation in soybean [14], we propose that GmCOP1s regulate plant height at least partially by regulating the abundance of STF1/2 in soybean.

### 2.5. GmCOP1a and GmCOP1b Regulate the Low Blue Light Response of Soybean

Given that GmCRY1s mediate low blue light (LBL)-induced shade avoidance syndrome (SAS), including exaggerated stem elongation in soybean [14], we tested if GmCOP1s are involved in LBL-induced SAS by analyzing the performance of indicated lines in response to LBL. The results revealed that the simultaneous knockout of *GmCOP1a* and *GmCOP1b* completely abolished the LBL-induced elongations of hypocotyls and epicotyls, confirming that GmCOP1s are required for LBL-induced SAS in soybean (Figure 5 and Appendix A). The efficiency of LBL in promoting stem elongation was significantly decreased with the absence of *GmCOP1b* (Figure 5), suggesting that knockout of the *GmCOP1b* gene may enhance the performance of soybean under high-density planting conditions.

### 2.6. Knockout of GmCOP1b Improves Soybean Performance under High-Density Condition

Next, we compared the agronomic traits between the wild-type TL1 and *Gmcop1b-2* mutant by field trails under 2 planting densities (30 cm and 15 cm plant space for normal and high densities, respectively) (Figure 6A). The results demonstrated that the TL1 soybeans showed higher plant height and significant reductions in node number, branch number, total grains per plant, and total grains weight per plant under the high-density condition compared to that under the normal density condition, suggesting that SAS essentially constrains the yield potential of TL1 cultivar in the field (Figure 6). Compared to TL1, the *Gmcop1b-2* plants were less sensitive to the change in planting density in the aspect of plant height, node number, total grains per plant, and total grains weight per plant (Figure 6 and Appendix A). As a consequence, the total grain weight of the *Gmcop1b-2* mutant was significantly higher than that of wild-type TL1 under high-density conditions. Taken together, these results demonstrated the potential value of utilizing the *Gmcop1b* mutant to breed high yield soybeans for density farming.

## 3. Discussion

COP1 is the key repressor of photomorphogenesis in higher plants. Given its indispensable roles in growth and development, the absence of COP1 in the model plant Arabidopsis results in lethality at embryo and seedling stages [38], and its partial loss-of-function mutants display constitutive photomorphogenic phenotypes in darkness, and extremely dwarf phenotype at seedling and adult stages, respectively [20,38]. The conserved functions of COP1 proteins across phylogenetic lines have also been evidenced by studies in other plants, including rice, pea, and tomato, but not in soybean previously [32,33,34,35,36,37]. Here, we revealed that GmCOP1s also play pivotal roles in repressing photomorphogenesis by promoting the degradation of bZIP transcriptional factor STF1/2 in soybean. In contrast to Arabidopsis, which harbors a single *COP1* gene, soybean carries two *COP1* orthologous genes with comparable but not exactly identical roles (Figure 1). We showed that the seedlings of the *Gmcop1a/b* double mutant, rather than that of the *Gmcop1a* or *Gmcop1b* single mutant, displayed constitutive photomorphogenesis in darkness, demonstrating that the two GmCOP1s are functionally redundant in mediating skotomorphogenesis in soybean (Figure 3D,E). Interestingly, the *Gmcop1a* mutant performed normally, while the *Gmcop1b* mutant was semi-dwarf under normal light conditions, suggesting that GmCOP1b plays a more prominent role than GmCOP1a in regulating plant height and other traits (Figure 4A,B and Appendix A). Taking the high identity between GmCOP1a and GmCOP1b (over 95%) together with the fact that both of them could equally rescue the *cop1–4* mutant phenotype in Arabidopsis, we surmise that the protein sequence variations are not the main cause of their functional diversification (Figure 1). As an alternative explanation, the transcriptional levels of *GmCOP1b* are relatively higher than that of *GmCOP1a* in most of the tested tissues, which is consistent with the hypothesis that gene expression divergence is an essential driving force for the functional evolution of duplicate genes (Figure 2) [39].

Although the functional mechanisms of COP1 have been extensively addressed in past decades, the utilization of COP1 in crop improvement has long been underestimated. Our previous study demonstrated that overexpression of GmCRY1b resulted in several ideal traits, including shorter internode and petiole length and more branches and pod numbers [14]. The *GmCRY1b-OX* lines were lodging resistant and dramatically enhanced the yield, suggesting a promising option for breeding high yield cultivars by elevating the GmCRY-mediated signal pathway. COP1 is the key repressor of the CRY-mediated photomorphogenesis pathway [16,17,28,29,30]. Therefore, negatively manipulating the activity of COP1 may confer similar ideal traits as overexpression of GmCRY1b. Here, we tested the performance of *Gmcop1b* mutant plants through field trials. As expected, the *Gmcop1* mutant plants performed similarly to the *GmCRY1b-OX* lines, characterized by short internode length, more effective branches and pod numbers, and higher yield per plant under high-density planting conditions ( Figure 6 and Appendix A). Overall, this study provides another piece of evidence that it is feasible to breed high yield soybean cultivars by manipulating the cryptochrome-mediated blue light signaling pathway.

## 4. Materials and Methods

### 4.1. Plant Materials and Growth Conditions

The Col-4 ecotype was used as a wild-type Arabidopsis. *YFP-GmCOP1a* and *YFP-GmCOP1b* lines were generated in the *cop1–4* background. Seeds of indicated lines were sterilized with 10% sodium hypochlorite solution, planted on 0.5 × MS medium with 10% sucrose, kept in the dark at 4 °C for 4 days, and then transferred to the darkness at 22 °C for 5 days to take photos and count the hypocotyl length, or transferred to continuous white light at 22 °C for 5 days for immunoblotting analysis.

The Tian Long 1 cultivar (TL1) was used as a wild type soybean for genetic transformation. The soybean cultivar used in this study was TL1 which was provided by the oil crops research institute, Chinese Academy of Agricultural Sciences. The *Gmcop1a-1*, *Gmcop1a-2*, *Gmcop1a-3*, *Gmcop1b-1*, *Gmcop1b-2*, *Gmcop1b-3* mutants were obtained by CRISPR-Cas9 technology, and the *Gmcop1a-2/1b-2* double mutant was obtained by crossing the *Gmcop1b-2* and *Gmcop1a-2* mutants. For expression pattern analysis, the Williams 82 (W82) soybean seedlings were planted under continuous light in an incubator at 26 °C for 14 days. The root, stem, cotyledon, unifoliolate leave, trifoliolate leave, and apex tissues were collected for RNA extraction. For the skotomorphogenic phenotype analysis of different lines, the plants were grown in the dark at 26 °C for 5 days. For the plant height phenotype analysis, different lines were planted under the short-day (SD, 12 h light/12 h dark) or long-day (LD, 16 h light/8 h dark) conditions at 26 °C for 10 days in an artificial climate greenhouse. For immunoblot analysis, the indicated lines were grown under SD conditions for 8 days, treated in darkness for 1 day, transferred to light, and sampled at 0 h, 2 h, and 4 h, respectively. To test the shade avoidance response to low blue light, different lines were planted under LD conditions at 26 °C for 3 days and then grown under low blue light conditions for 7 days. For the investigation of agronomic traits, TL1 and *Gmcop1b-2* were planted in Beijing (40.1 N, 116.7 E). The plant height, node number, branch number, total grains per plant, and total grain weight per plant of indicated lines were measured at the mature stage, respectively.

### 4.2. Plasmid Construction and Genetic Transformation

To generate the *YFP-GmCOP1a* and *YFP-GmCOP1b* overexpression lines, the CDS sequences of *GmCOP1a* (*Glyma.02G267800*) and *GmCOP1b* (*Glyma.14G049700*) were amplified from the cDNA of W82, cloned into the entry vector pDONR (Zeo) through BP reaction, and then cloned into the destination binary vector pearleygate104 through LR reaction by Gateway system (Invitrogen, Waltham, MA, USA), respectively. The constructs were introduced into *Agrobacterium tumefaciens* strain AGL0 by electroporation and transformed into *cop1–4* by the inflorescence soaking method [40].

To generate CRISPR/Cas9-engineered mutants, the gRNAs were designed by the CRISPR-P website (http://cbi.hzau.edu.cn/CRISPR2/, accessed on 1 February 2017) and constructed the plasmids according to the method described before [41]. Briefly, the DNA fragment for GmU6 was amplified with the primer pairs GmU6-F and GmU6-R; the DNA fragment for sgRNA was amplified with primer pairs sgRNA-F and Scaffold-R; the GmU6-sgRNA fragment was amplified with the primer GmU6-F and scaffold-R by overlapping PCR and then inserted into the 35S-CAS9 vector through XbaI site. The corresponding gRNA sequences used were as follows: g1 (CCGCCGTCGTCAACCTGAACCG) and g2 (TTGCAGATGTTGACGGTTCTGG) for *GmCOP1a*, g3 (TTACGGATGCTTTGACGACTCTGG) and g4 (ACTTCATTAGTGCTGTATGCTGCTGG) for *GmCOP1b.* The CRISPR-CAS9 plasmids were introduced into Agrobacterium tumefaciens strain EHA105 by electroporation and transformed into soybean TL1 by the cotyledon-node method [42]. The primer sequences used in the construction of the above plasmids are shown in Appendix A Appendix A. The PCR amplification was performed as follows: pre-denaturation at 98 °C for 2 min, followed by a 35-cycle program (98 °C for 10 s, 57 °C for 5 s, and 72 °C for 1 Kb/min), and final extension at 72 °C for 7 min.

### 4.3. Genotyping of Gene Editing Mutants

The T0 generation plants were sprayed with 100 mg/L of glufosinate–ammonium solutions. Genomic DNA of herbicide-resistant plants was extracted by the CTAB method, and the herbicide-resistant plants were further confirmed by identifying the existence of BAR, CAS9, and GmU6-sgRNA fragments with specific primers. The DNA fragment of the target site was amplified by PCR for DNA sequencing to determine homozygous mutants in the offspring of transgenic lines. The primer sequences used for genotyping are displayed in Appendix A Appendix A. The PCR amplification was performed as follows: pre-denaturation at 95 °C for 30 s, followed by a 35-cycle program (95 °C for 30 s, 57 °C for 30 s, and 72 °C for 1 Kb/min), and final extension at 72 °C for 5 min.

### 4.4. Phylogenetic Tree Analysis

The amino acid sequences of COP1s in different species (*Arabidopsis*, soybean, tomato, *Brachypodium*, rice, corn, wheat, sorghum, millet, mouse, and human) were acquired from Phytozome (https://phytozome.jgi.doe.gov/pz/portal.html, accessed on 10 January 2017) and NCBI (https://www.ncbi.nlm.nih.gov/, accessed on 10 January 2017) and aligned using ClustalW. The phylogenetic tree was constructed using a neighbor-joining method by MEGA7 with default parameters [43]. The bootstraps value was set to 1000.

### 4.5. Gene Expression Analysis

Different tissues were frozen in liquid nitrogen and ground using a mortar. Total RNA was extracted by the Trizol reagent (TIANGEN, Beijing, China). Then, 3 μg of total RNA was used for the first-strand cDNA synthesis by the TransScript II First-Strand cDNA Synthesis SuperMix (TransGen, Beijing, China) following the manufacturer’s instructions. The cDNAs were diluted 10-fold for quantitative PCR (qPCR) using the SYBR premix ex (TAKARA, Dalian, China) on a Roche LightCycler 480 System. *GmACT11* (*Glyma.18G290800*) was used as reference genes. Three biological replicates were performed for each experiment. The qPCR amplification was performed as follows: pre-denaturation at 95 °C for 30 s, followed by a 40-cycle program (95 °C for 5 s and 60 °C for 20 s).

### 4.6. Immunoblot Analysis

Immunoblots were performed as previously described [44]. The NC membranes of YFP-GmCOP1a and YFP-GmCOP1b lines were immunoblotted with anti-GFP antibody (MBL, Beijing, China), stripped, and re-probed with anti-HSP antibody as an internal control. For analysis of STF1/2 protein in indicated lines, immunoblots were performed using the anti-STF2 antibody [14]. The protein bands were quantified by Image J software as described before [44].

## Figures and Tables

**Figure 1 ijms-23-05394-f001:**
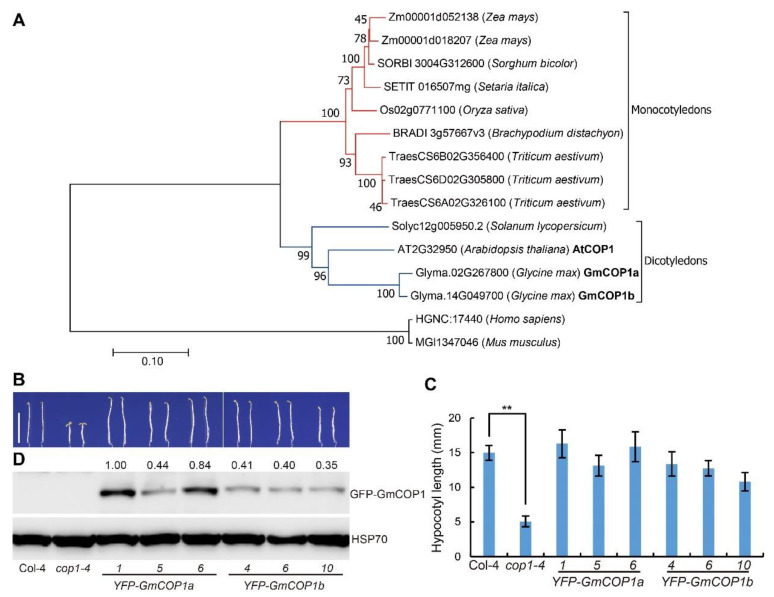
Ectopic expression of *GmCOP1a* or *GmCOP1b* in the *Arabidopsis cop1–4* mutant. (**A**) Phylogenetic analysis of COP1 proteins from indicated species. The sequences of COP1 homologous proteins were aligned by ClustalW to construct the neighbor-joining phylogenetic tree with 1000 bootstrap replicates by MEGA7. The two soybean homologous proteins, GmCOP1a (Glyma.02G267800) and GmCOP1b (Glyma.14G049700), as well as AtCOP1, are highlighted in bold. The red lines indicate the monocotyledons; the blue lines indicate dicotyledons. (**B**) Hypocotyl phenotypes of the *YFP-GmCOP1a* and *YFP-GmCOP1b* transgenic lines in the *cop1–4* mutant background. Seedlings were grown in darkness for 5 days, scale bar = 1 cm. (**C**) Statistical analysis of hypocotyl lengths in (**B**). The values represent the means ± SD (*n* ≥ 15). The significant difference was determined by Student’s *t*-test (** *p* < 0.01). (**D**) Immunoblot shows the expression levels of YFP-GmCOP1a or YFP-GmCOP1b proteins in the indicated lines using the anti-GFP antibody. HSP70 was used as a reference protein. The numbers at the top indicate the relative abundance of transgenic proteins, which was calculated by the formula GFP/HSP70.

**Figure 2 ijms-23-05394-f002:**
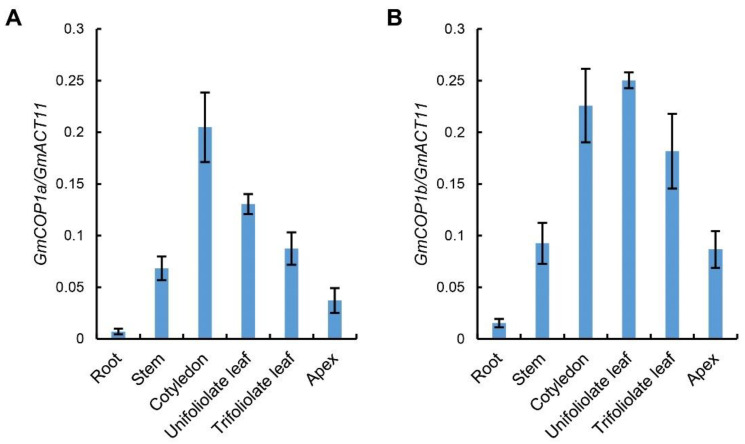
Expression analysis of *GmCOP1a* and *GmCOP1b* in different tissues of soybean. (**A**,**B**) Quantitative PCR (qPCR) results showing the *GmCOP1a* (**A**) and *GmCOP1b* (**B**) mRNA levels in the indicated tissues. The values represent the means ± SD (*n* = 3). *GmACT11* is used as a reference gene.

**Figure 3 ijms-23-05394-f003:**
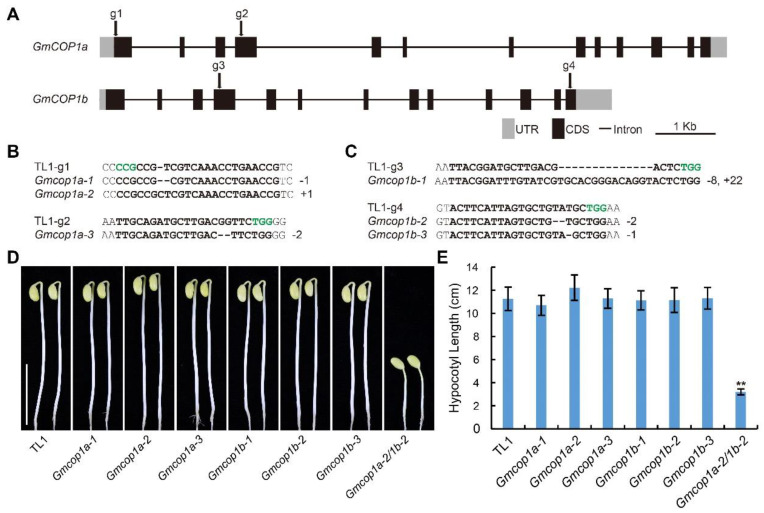
Generation and characterization of the CRISPR-Cas9 engineered *Gmcop1a*, *Gmcop1b,* and *Gmcop1a1b* mutants. (**A**) Schematic diagram showing the genomic structures of indicated genes and the gRNA targeting sites (g1 to g4). Dark boxes represent coding sequence (CDS). Grey boxes denote untranslated region (UTR). Dark lines indicate intergenic region (Intron). scale bar = 1 Kb. (**B**,**C**) Alignment of the DNA sequences around the gRNA targeting sites in the wild type TL1, *Gmcop1a* mutants (**B**) and *Gmcop1b* mutants (**C**). The gRNA targets and PAM sequences are highlighted in bold and green font, respectively. (**D**) Representative images of the indicated lines grown in darkness for 5 days. Scale bar = 10 cm. (**E**) Statistical analysis of hypocotyl length of different lines in (**D**). The values represent the means ± SD (*n* ≥ 9). Asterisks indicate significant difference compared to TL1 by Student’s *t*-test (** *p* < 0.01).

**Figure 4 ijms-23-05394-f004:**
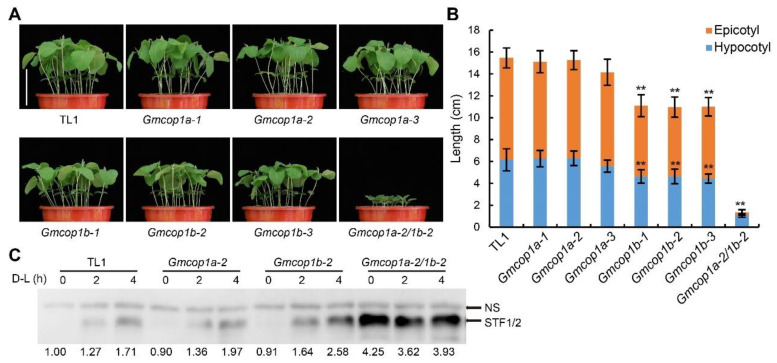
Comparisons of seedling growth and STF1/2 protein accumulation among different GmCOP1 mutant lines. (**A**) Representative images of indicated lines grown under short-day (12 h light/12 h dark) conditions for 10 days. Scale bar = 10 cm. (**B**) Statistical analysis of hypocotyl and epicotyl lengths in (**A**). The values represent the means ± SD (*n* ≥ 9); asterisks indicate significant differences compared to TL1 by Student’s *t*-test (** *p* < 0.01). (**C**) Immunoblot showing the STF protein abundances in the indicated lines. NS is a non-specific band used as loading control. Numbers at the bottom represent the ratios of STF1/2 to NS. Soybean plants were grown under 12 h light/12 h dark conditions for 8 days, kept in darkness for 1 day, then exposed to white light, and sampled at 0 h, 2 h, and 4 h, respectively.

**Figure 5 ijms-23-05394-f005:**
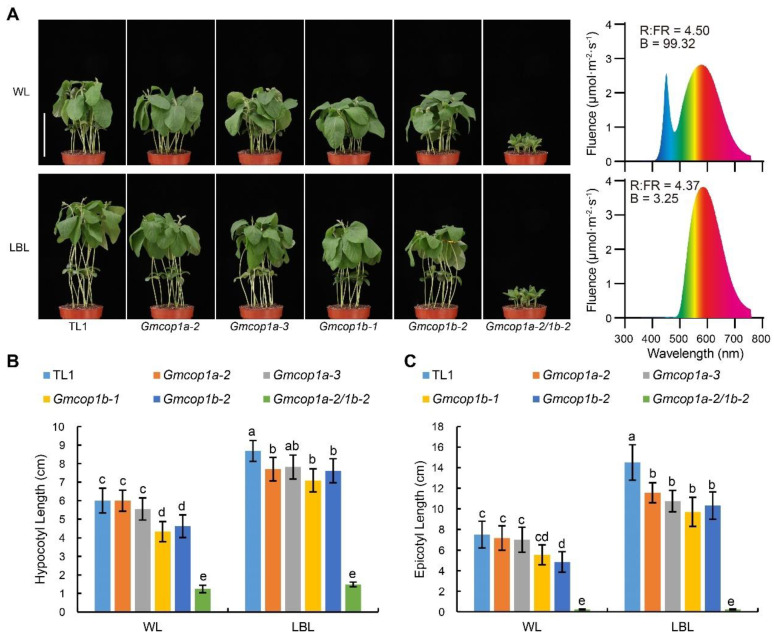
Comparisons of LBL-induced stem elongation among different *GmCOP1s* mutant lines. (**A**) Left panel: Representative images of indicated lines subjected to white light (WL) or LBL treatment. Right panel: the spectrum of WL and LBL. The intensity of photosynthetically active radiation (PAR) (400–700 nm) of WL or LB is 500 μmol·m^−2^·S^−1^. Seedlings germinated using white light under long-day (16 h light/8 h dark) conditions for 3 days, then grown under long-day conditions with the indicated light for 7 days. Scale bar = 10 cm. (**B**,**C**) Statistical analysis of hypocotyl length (**B**) and epicotyl length (**C**) of the seedlings in (**A**). The values represent the means ± SD (*n* ≥ 8). The lowercase letters indicate significant differences as determined by two-way ANOVA with Tukey’s post hoc test (*p* < 0.01).

**Figure 6 ijms-23-05394-f006:**
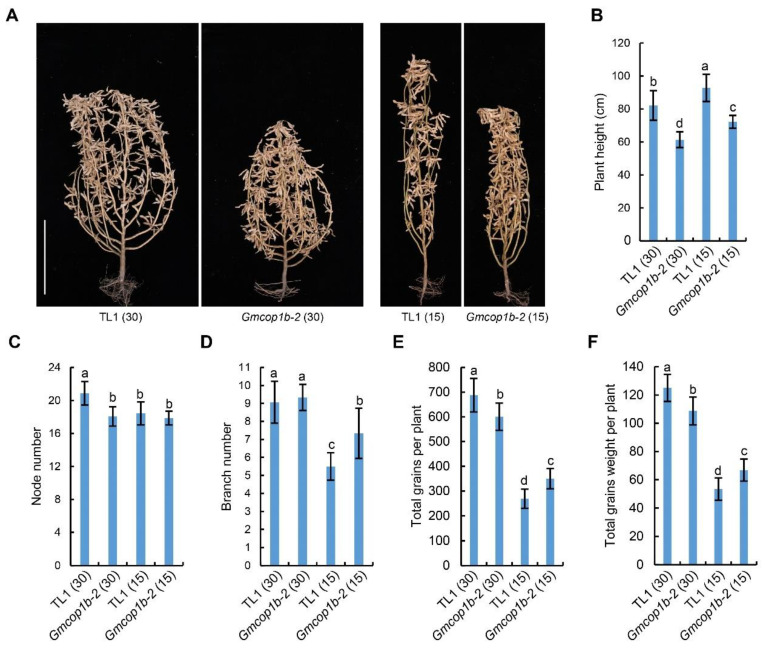
Agronomic traits of the *Gmcop1b-2* mutant and TL1 under different planting densities in the field. (**A**) Representative images of the indicated lines grown with a plant spacing of 30 cm or 15 cm at the R8 stage. Scale bar = 30 cm. (**B**–**F**) Statistical analysis of plant height (**B**), node number (**C**), branch number (**D**), total grains per plant (**E**), and total grains weight per plant (**F**) of the lines in (**A**). The lowercase letters indicate significant differences as determined by two-way ANOVA with Tukey’s post hoc test (*p* < 0.01). Numbers in the brackets indicate the plant spacing, and the unit is cm.

## Data Availability

The data presented in this study are available on request from the corresponding author.

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
