# Peer review of "Induced Mutation in *GmCOP1b* Enhances the Performance of Soybean under Dense Planting Conditions"

_ijms, 2022, doi:10.3390/ijms23105394_

Round 1

Reviewer 1 Report

Ji et al revealed the functional role of COP1 knockout by using CRISPR CAS9 in soybean. The findings are very interesting. However, some suggestions are enlisted below. 

1)  Next time, insert the line number in MS, so it will be easy to comments

2)  Abstract: clearly mention the mutant (e.g. INDELS) using CRISPR.

3) Introduction:  Although the functions of COP1 in light signaling have ..... elaborate the function of COP1 from these reference studies.

4) Results:  This is possibly because the
GmCOP1s was driven by the 35S promoter instead of the native COP1 promoter ?...... how did the author can assume its role? any experimental evidence or previous reports or the author have tested the expression of some genes regarding leaf size?

5) Draw a model figure that is the representation of Gene Cascades Involved in  COP1 functions.....

Author Response

please see the attachment, thank you for your review.

Reviewer 2 Report

Ji et al.....Manuscript was in good agreement with the findings of the investigation. There are a few typos and grammatical flaws that will, sadly, detract from readers' perceptions of the article's quality. Before re-submitting the paper, I would advise the authors to have it thoroughly examined and proofread. I also have a few minor suggestions for you to consider.

1) Results: we searched the soybean genomic sequence...based on keywords or using Arabidopsis gene

2) we made the YFP is yellow fluorescent protein?? mention it at least in one place

3) Notably, although....avoid these two, rephrase it

4) In Materials and methods, seeds sterilized with 10% bleach, give the material names rather than common one. 

5) Please include the PCR conditions methods

Author Response

(The authors gave the same response as above.)
